# The Lyme disease agent co-opts adiponectin receptor-mediated signaling in its arthropod vector

Xiaotian Tang[1]*[†], Yongguo Cao[1,2][†], Gunjan Arora[1], Jesse Hwang[1], Andaleeb Sajid[1], Courtney L Brown[3], Sameet Mehta[4], Alejandro Marín-López[1], Yu-Min Chuang[1], Ming-Jie Wu[1], Hongwei Ma[1,5], Utpal Pal[6], Sukanya Narasimhan[1], Erol Fikrig[1]*

[1]Section of Infectious Diseases, Department of Internal Medicine, School of Medicine, Yale University, New Haven, United States; [2]Department of Clinical Veterinary Medicine, and Key Laboratory for Zoonosis Research, Ministry of Education, College of Veterinary Medicine, Jilin University, Changchun, China; [3]Yale Combined Program in the Biological and Biomedical Sciences, Yale University, New Haven, United States; [4]Yale Center for Genome Analysis, Yale University, New Haven, United States; [5]Department of Microbiology, School of Basic Medicine, Fourth Military Medical University, Shaanxi, China; [6]Department of Veterinary Medicine, University of Maryland, College Park, College Park, United States

*For correspondence:
xiaotian.tang@yale.edu (XT);
erol.fikrig@yale.edu (EF)

[†]These authors contributed equally to this work

Competing interest: The authors declare that no competing interests exist.

**Abstract** Adiponectin-mediated pathways contribute to mammalian homeostasis; however, little is known about adiponectin and adiponectin receptor signaling in arthropods. In this study, we demonstrate that *Ixodes scapularis* ticks have an adiponectin receptor-like protein (ISARL) but lack adiponectin, suggesting activation by alternative pathways. *ISARL* expression is significantly upregulated in the tick gut after *Borrelia burgdorferi* infection, suggesting that ISARL signaling may be co-opted by the Lyme disease agent. Consistent with this, RNA interference (RNAi)-mediated silencing of *ISARL* significantly reduced the *B. burgdorferi* burden in the tick. RNA-seq-based transcriptomics and RNAi assays demonstrate that ISARL-mediated phospholipid metabolism by phosphatidylserine synthase I is associated with *B. burgdorferi* survival. Furthermore, the tick complement C1q-like protein 3 interacts with ISARL, and *B. burgdorferi* facilitates this process. This study identifies a new tick metabolic pathway that is connected to the life cycle of the Lyme disease spirochete.

## Editor's evaluation

This work is a superb demonstration of how *B. burgdorferi* hijacks the ISARL-mediated phospholipid metabolism pathway to facilitate survival inside *Ixodes scapularis* ticks. Unexpectedly, the authors show that *B. burgdorferi* upregulates the tick complement C1q-like protein 3, which interacts with ISARL, to provide the required metabolic needs for the bacterium.

## Introduction

Adiponectin, adipocyte complement-related protein of 30 kDa (or Acrp30), plays important roles in the regulation of metabolism, insulin sensitivity, and inflammation across species (*Kadowaki et al., 2006*; *Ouchi and Walsh, 2007*; *Yamauchi et al., 2002*). Adiponectin mediates its actions mainly via binding adiponectin receptors with its globular C1q domain (*Buechler et al., 2010*; *Yamauchi et al., 2002*). Two adiponectin receptors, AdipoR1 and AdipoR2, have been identified

**eLife digest** Many countries around the world are seeing an increase in the number of patients diagnosed with Lyme disease, with often serious joint, heart, and neurologic complications. This illness is caused by species of 'spirochete' bacteria that live and multiply inside black-legged ticks, and get injected into mammals upon a bite. Ticks are not simply 'syringes' however, and a complex relationship is established between spirochetes and their host. This is particularly true since Lyme disease-causing bacteria such as *Borrelia burgdorferi* rely on ticks to obtain energy and nutrients.

Tang, Cao et al. delved into these complex interactions by focusing on the molecular cascades (or pathways) involving adiponectin, a hormone essential for regulating sugar levels and processing fats. Analyses of gene and protein databases highlighted that ticks carry a receptor-like protein for adiponectin but not the hormone itself, suggesting that an alternative pathway is at play. This may involve *B. burgdorferi*, which gets its fats and sugars from its host.

And indeed, experiments showed that ticks produced more of the adiponectin receptor-like protein when they carried *B. burgdorferi*; conversely, silencing the receptor reduced the number of surviving spirochetes inside the tick. Further exploration showed that the receptor mediates molecular cascades that help to process fat molecules; these are associated with spirochete survival. In addition, the receptor-like protein was activated by C1QL3, a 'complement 1q domain-contained' molecule which might be part of the tick energy-making or immune systems. Larger quantities of C1QL3 were found in ticks upon *B. burgdorferi* infection, suggesting that the spirochete facilitates an interaction that boosts activity of the adiponectin receptor-like protein.

Overall, the work by Tang and Cao et al. revealed a new pathway which *B. burgdorferi* takes advantage of to infect their host and multiply. Targeting this molecular cascade could help to interfere with the life cycle of the spirochete, as well as fight Lyme disease and other insect-borne conditions.

in mammals (*Yamauchi et al., 2003*). AdipoR1 and R2 belong to a family of membrane receptors predicted to contain seven transmembrane (TM) domains with an internal N terminus and an external C terminus (*Yamauchi et al., 2003*). AdipoR1 has a higher binding affinity for the globular form of adiponectin, whereas AdipoR2 has a greater affinity for full-length adiponectin (*Yamauchi et al., 2003*). Interestingly, AdipoR1 and AdipoR2 double-knockout mice have increased triglyceride levels and exhibit insulin resistance, demonstrating that AdipoR1 and AdipoR2 regulate lipid and glucose homeostasis (*Kadowaki et al., 2006*; *Yamauchi et al., 2007*). In yeast, a homolog of mammalian adiponectin receptors, ORE20/PHO36, is involved in lipid and phosphate metabolism (*Karpichev et al., 2002*). PHO36 can also interact with a plant protein, osmotin, a homolog of mammalian adiponectin, thereby controlling apoptosis in yeast (*Narasimhan et al., 2005*). Adiponectin and adiponectin receptors in disease-transmitting arthropods have not been characterized. By utilizing the amino acid sequence homology search in other model arthropods, adiponectin was not identified from *Drosophila melanogaster*; however, an adiponectin receptor that regulates insulin secretion and controls glucose and lipid metabolism was characterized (*Kwak et al., 2013*). In addition, *Zhu et al., 2008* cloned an adiponectin-like receptor gene from the silk moth, *Bombyx mori*, and found that infection with *B. mori* nucleopolyhedrovirus significantly increased adiponectin receptor mRNA levels in the midgut of susceptible *B. mori*, suggesting an association with pathogen infectivity.

*Ixodes scapularis*, the black-legged tick, is an important vector of the Lyme disease agent, *Borrelia burgdorferi* (*Estrada-Peña and Jongejan, 1999*), which causes approximately 300,000 cases annually in the United States (*Rosenberg et al., 2018*). *B. burgdorferi* is acquired when larval or nymphal ticks feed on infected animals, and is transmitted by nymphs or adults to vertebrate hosts (*Kurokawa et al., 2020*). Lyme disease in humans manifests as a multisystem disorder of the skin and other organs (e.g., joints, heart, and nervous system), resulting in patients experiencing cardiac, neurological, and arthritic complications (*Asch et al., 1994*; *Singh and Girschick, 2004*). A human vaccine against Lyme disease was approved by the FDA but is not currently available (*Steere et al., 1998*). Targeting tick proteins has the potential to disrupt tick feeding and alter *B. burgdorferi* colonization or transmission (*Kurokawa et al., 2020*), thereby offering a new way to interfere with the life cycle of the Lyme disease spirochete.

In the present study, we demonstrate that an *I. scapularis* adiponectin receptor-like (ISARL) protein facilitates *B. burgdorferi* colonization of the tick. ISARL-mediated stimulation of *I. scapularis* metabolic pathways are associated with spirochete colonization, and a tick complement C1q-like protein 3 contributes to ISARL activation.

## Results

### Identification and characterization of an *I. scapularis* adiponectin receptor-like protein

As tick metabolism changes during pathogen colonization, and adiponectin-associated pathways mediate diverse metabolic activities, we examined the *I. scapularis* database for two of the prominent genes linked to this pathway. The available *I. scapularis* database (taxid:6945) in NCBI was searched with the genes for mammalian adiponectin and adiponectin receptors, and results with the human and mouse genes are shown. There were no tick genes with high homology to the genes for human and mouse adiponectin full-length sequences. Interestingly, there was an *I. scapularis* gene (GenBank number: XM_029975213) with substantial homology to the human and murine adiponectin receptors, which we designated *I. scapularis* adiponectin receptor-like (*ISARL*). The corresponding ISARL protein sequence (GenBank number: XP_029831073) was also identified. The full-length *ISARL* mRNA encoded a protein with 384 amino acid residues and 71% amino acid sequence similarity to both the human and mouse adiponectin receptor proteins 1 and 2. It also has high similarity (87%) to homologs from insect species, including the *D. melanogaster* adiponectin receptor (GenBank number: NP_732759) (*Figure 1—figure supplement 1*). Structure prediction and hydrophobicity analysis indicated that ISARL has seven TM domains (*Figure 1—figure supplement 2*). Comparison of the amino acid sequences between vertebrate and invertebrate species revealed that the predicted TM regions are highly conserved, especially in the TM3 domain (*Figure 1—figure supplement 1*).

### Silencing *ISARL* reduces *B. burgdorferi* colonization by *I. scapularis* nymphs

As *I. scapularis* lack an obvious adiponectin homolog, we examined whether expression of *ISARL* could be stimulated in the feeding vector by allowing ticks to engorge on mice, including uninfected and *B. burgdorferi*-infected animals. Interestingly, a blood meal containing *B. burgdorferi* resulted in significantly increased expression of *ISARL* in the nymphal tick guts (p<0.0001) (*Figure 1A*). This suggests that the presence of *B. burgdorferi* in the blood meal helps to stimulate tick metabolic activity and/or that ISARL may have an important role during *B. burgdorferi* colonization of the tick gut.

Since *ISARL* expression was upregulated upon *B. burgdorferi* infection, we hypothesized that RNAi-mediated silencing of *ISARL* would affect *B. burgdorferi* colonization by nymphal *I. scapularis*. To this end, *ISARL* or *GFP* (control) dsRNA was injected into the guts of pathogen-free nymphs by anal pore injection. Then, the ticks were allowed to feed on *B. burgdorferi*-infected mice. Quantitative RT-PCR (qPCR) analysis showed a significant decrease of *ISARL* expression in the guts of ds *ISARL*-injected ticks (p<0.01) when compared to that in control ds *GFP*-injected tick guts (*Figure 1B*), indicating that the knockdown was successful. The engorgement weights of ds *ISARL*-injected nymphs and control ds *GFP*-injected nymphs were comparable (p>0.05) (*Figure 1C*), suggesting that silencing *ISARL* had no effect on tick feeding behavior. However, *ISARL*-silenced nymph guts showed a marked reduction of the *B. burgdorferi* burden (p<0.001) when compared to that in control ticks (*Figure 1D*), demonstrating that ISARL is associated with *B. burgdorferi* colonization in the nymphal tick gut.

### Silencing *ISARL* does not affect *B. burgdorferi* transmission by *I. scapularis* nymphs

To determine whether ISARL might also play a role in the transmission of *B. burgdorferi* to the mammalian host, we silenced *ISARL* in *B. burgdorferi*-infected nymphs by microinjection of ds *ISARL* into the ticks. The results showed that *B. burgdorferi* burdens in the skin of mice (ear skin distal from the tick bite site) at 7, 14, and 21 days post tick detachment, and in heart and joint tissues 21 days post tick detachment were comparable (p>0.05) in mice fed upon by ds *GFP*- or by ds *ISARL*-injected nymphs (*Figure 1E and F*), suggesting that silencing *ISARL* had no observable effect on *B. burgdorferi* transmission by *I. scapularis* nymphs.

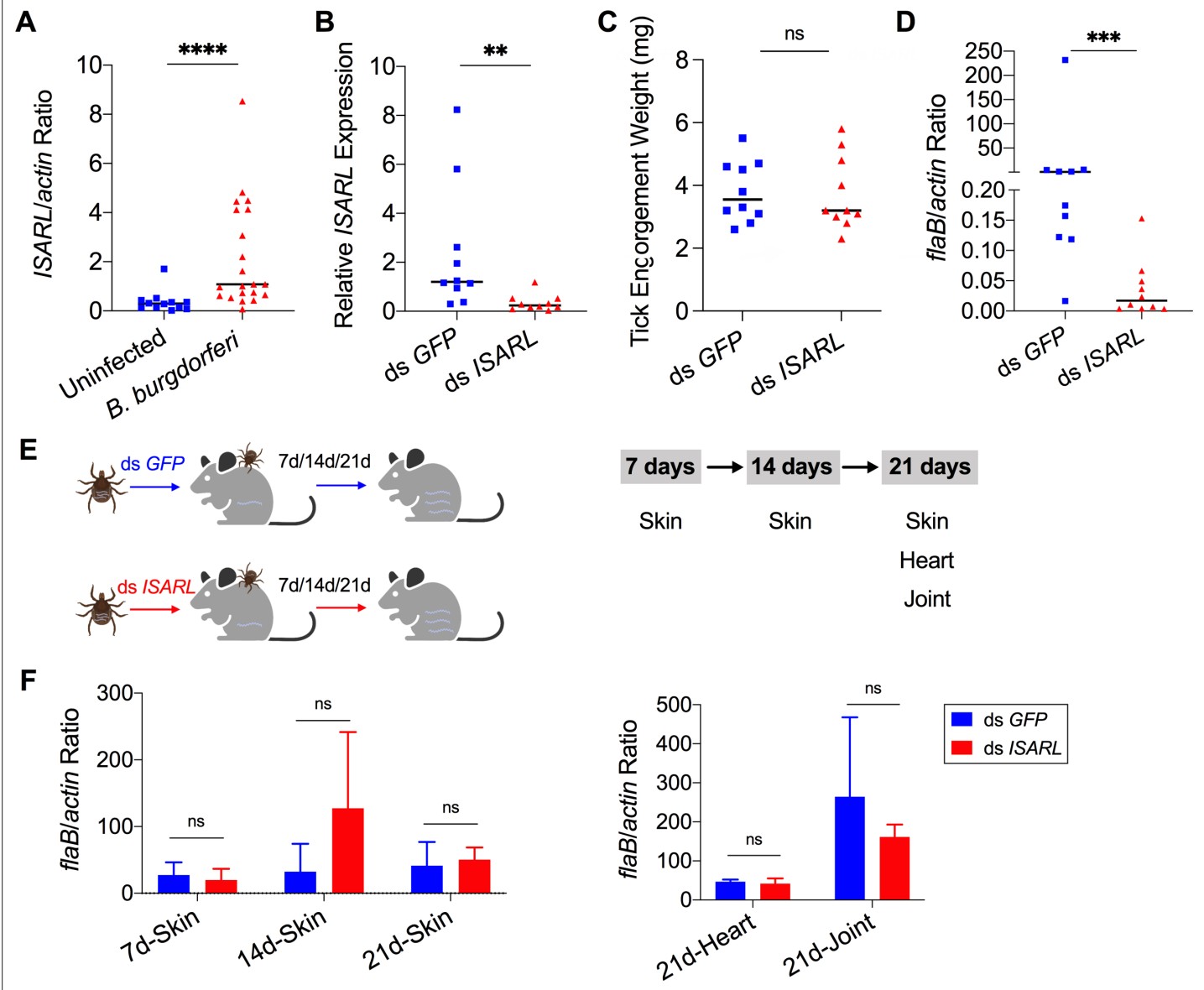

**Figure 1.** Silencing of *ISARL* significantly reduces the *B. burgdorferi* burden in nymphal tick guts. (**A**) *ISARL* is significantly induced in nymphal tick guts after feeding on *B. burgdorferi*-infected mice. (**B**) qPCR assessment of *ISARL* transcript levels following RNAi silencing of *ISARL* after feeding on *B. burgdorferi*-infected mice. (**C**) Nymphal engorgement weights in *ISARL*-silenced and mock-injected nymphs. Each data point represents one engorged tick. (**D**) qPCR assessment of *B. burgdorferi flaB* levels in guts following RNAi silencing of *ISARL* after feeding on *B. burgdorferi*-infected mice. Each data point represents one nymph gut. Horizontal bars in the above figures represent the median. Statistical significance was assessed using a nonparametric Mann–Whitney test (ns, p>0.05; **p<0.01; ***p<0.001; ****p<0.0001). (**E**) *Borrelia*-infected nymphs microinjected with ds *ISARL* or ds *GFP* were fed on clean mice to assess transmission of the spirochete. The infection of *Borrelia* in murine skin 7, 14, and 21 days after infection, and in heart and joint tissues at 21 days was determined. (**F**) Murine skin 7, 14, and 21 days after infection, and in heart and joint tissues at 21 days was determined by qPCR of *flaB* and normalized to mouse *actin*. Data represent the means ± standard deviations from five biological replicates with two technical replicates.

The online version of this article includes the following source data and figure supplement(s) for figure 1:

**Source data 1.** ISARL is involved in *B. burgdorferi* colonization in nymphal tick guts but has no effect on transmission.

**Figure supplement 1.** Protein sequence comparison of adiponectin receptors.

**Figure supplement 2.** Predicted protein structure and hydrophobicity of ISARL.

## Potential ISARL-dependent pathways associated with *B. burgdorferi* colonization

To investigate the mechanisms underlying the association of ISARL with *B. burgdorferi* colonization by *I. scapularis*, we assessed the presence or absence of ISARL on tick physiology by comparing transcriptomes of ds *ISARL* and ds *GFP* (control)-injected ticks after engorgement on *B. burgdorferi*-infected or uninfected mice using RNA-seq.

After feeding on uninfected mice, 18 genes were significantly differentially expressed in the guts of ds *ISARL*-injected nymphal ticks when compared to that in control ds *GFP*-injected tick guts (*Figure 2A*; *Supplementary file 1*), while 35 genes were differentially expressed after feeding on *B. burgdorferi*-infected mice (*Figure 2B*; *Supplementary file 1b*). In particular, the *ISARL* gene was successfully silenced by RNAi as demonstrated by transcriptome analysis (*Supplementary file 1*) and qPCR validation (*Figure 2C*). No common genes except *ISARL* were observed between ticks feeding on uninfected or *B. burgdorferi*-infected mice (*Figure 2D*), suggesting that the 34 genes (*Figure 2B*; *Supplementary file 1*) were all altered by *B. burgdorferi*, or the influence of *B. burgdorferi* on the host blood components, rather than blood meal itself, in the absence of ISARL.

In response to the blood meal, a significant change of the metabolic pathways in ticks was observed in the absence of ISARL. In particular, based on Gene Ontology (GO) functional classification and Kyoto Encyclopedia of Genes and Genomes (KEGG) pathways analyses, glutathione metabolism, including six gamma glutamyl transpeptidase genes (*Supplementary file 1*), was significantly altered in the absence of ISARL after engorgement of ticks on uninfected mice.

Similarly, many metabolism-associated genes were significantly downregulated in the absence of ISARL after engorging on *B. burgdorferi*-infected mice (*Supplementary file 1*). GO functional classification and KEGG pathways also showed that the most downregulated genes were involved in fatty acid (e.g., 3-hydroxyacyl-CoA dehydrogenase), lipid and phospholipid (e.g., phosphatidylserine synthase I), and purine (e.g., GMP synthase) metabolism pathways after silencing *ISARL* (*Figure 2E*), suggesting that ISARL functions as a metabolic moderator in ticks.

## ISARL-mediated phospholipid metabolic pathways affect *B. burgdorferi* colonization

To further investigate the exact metabolism pathway(s) involved in *B. burgdorferi* colonization, we first selected 18 well-annotated and metabolism-related differentially expressed genes to validate the accuracy and reproducibility of the transcriptome bioinformatic analyses by qPCR. The samples for qPCR validation are independent of the sequencing samples. In general, the qPCR results indicated that all the tested genes showed concordant direction of change with the RNA-seq bioinformatic data except one gene, pyridoxine kinase (*PDXK*) (*Figure 2—figure supplement 1*), indicating the accuracy and reliability of our RNA-seq libraries. Of these 17 downregulated genes, 4 genes showed significant downregulation profiles (p<0.05). These four genes included phosphatidylserine synthase I (*PTDSS1*) (*Figure 2F and G*), N-CAM Ig domain-containing protein (*NCAM*), vacuolar H+-ATPase V1 sector, subunit G (*V-ATPase*), and sideroflexin 1,2,3, putative (*SFXN*) (*Figure 2—figure supplement 2*).

Then, we silenced these four genes individually and investigated their potential roles in *B. burgdorferi* colonization. We also silenced another four genes, whose p-values were very close to significant (*Figure 2—figure supplement 2*). These four genes included 3-hydroxyacyl-CoA dehydrogenase, putative (*3HADH*), adenylosuccinate synthetase (*ADSS*), GMP synthase, putative (*GMPS*), and alpha-actinin, putative (*ACTN*). We did not observe a significant decrease of *B. burgdorferi* burden in nymphal tick guts after silencing *NCAM*, *V-ATPase*, *SFXN*, *ADSS*, *GMPS*, and *ACTN* compared to ds *GFP*-injected ticks (*Figure 2—figure supplement 3*). Instead, we found that *PTDSS1*-silenced nymphs showed a marked reduction in the *B. burgdorferi* burden in the guts when compared to that in control ticks (p<0.05) (*Figure 2H*). Furthermore, a blood meal containing *B. burgdorferi* resulted in significantly increased expression of *PTDSS1* in the nymphal tick guts (p<0.05) (*Figure 2I*), suggesting that PTDSS1 indeed has a critical role during *B. burgdorferi* colonization of the tick gut. PTDSS1 is involved in phospholipid metabolism and mainly uses L-serine as the phosphatidyl acceptor to generate the anionic lipid phosphatidylserine (PS), which serves as a precursor for phosphatidylethanolamine (PE) and phosphatidylcholine (PC) synthesis (*Figure 2J*; *Aktas et al., 2014*). Importantly, PC is one of the main phospholipids on the cellular membrane of *B. burgdorferi* (*Kerstholt et al., 2020*). However, *B. burgdorferi* lacks the central phospholipid metabolic enzymes. To further validate that

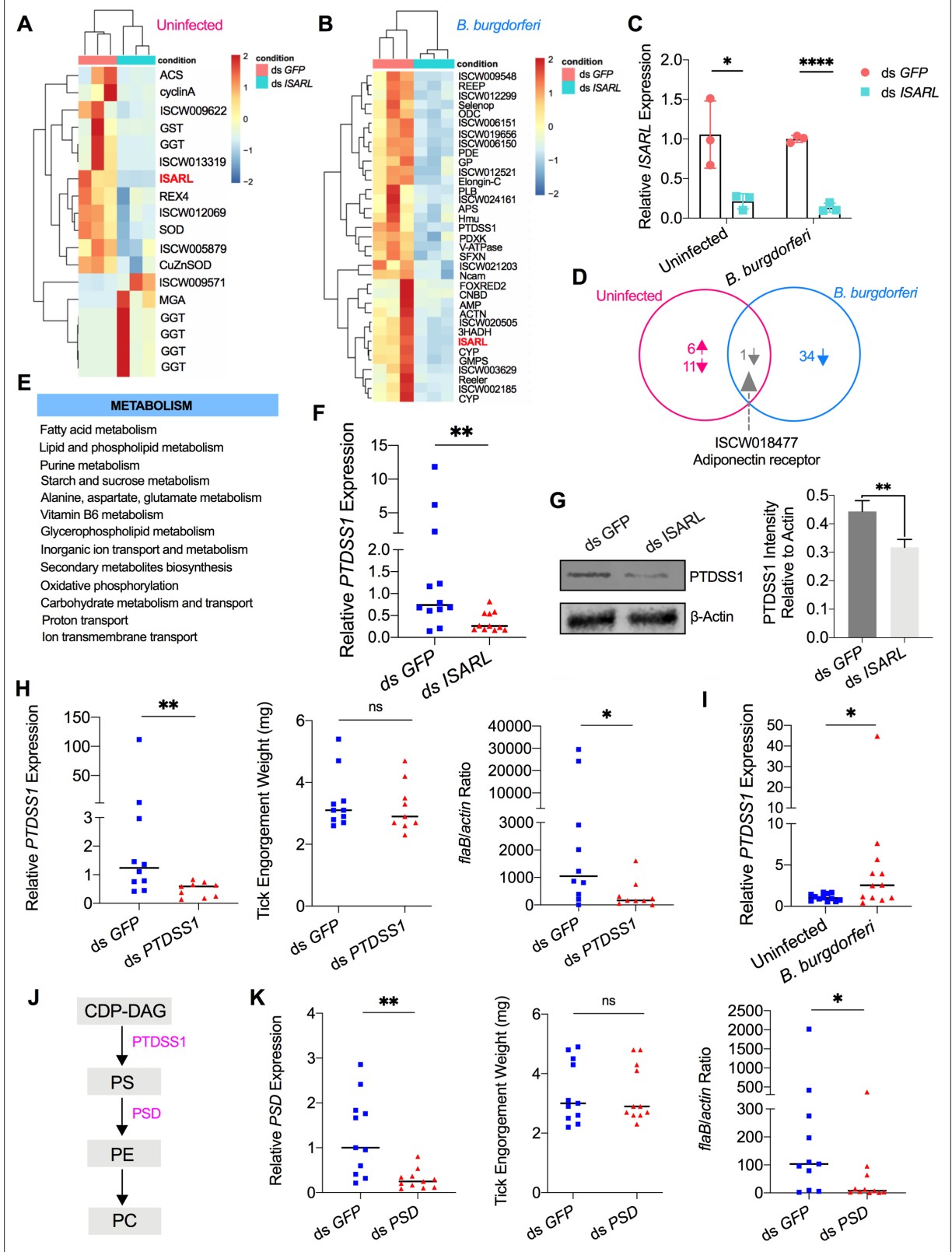

**Figure 2.** RNA-seq, qPCR validation, and RNAi-silencing assays revealed that phosphatidylserine synthase 1 (*PTDSS1*) is regulated by ISARL and is involved in *B. burgdorferi* colonization. (**A**) Hierarchical clustering of differentially expressed genes was generated after feeding on clean mice. (**B**) Hierarchical clustering of differentially expressed genes was generated after feeding on *B. burgdorferi*-infected mice. Each column represents biological replicates. The gene names can be found in ***Supplementary file 1***. The *ISARL* gene is highlighted with bold and red color. The expression

*Figure 2 continued on next page*

*Figure 2 continued*

levels were visualized, and the scale from least abundant to highest range is from –2.0 to 2.0. The phylogenetic relationships of differentially expressed genes are shown on the left tree. The top tree indicates the cluster relationship of the sequenced samples. (**C**) qPCR validation of *ISARL* knockdown in tick gut. Statistical significance was assessed using Student's *t* test (*p<0.05; ****p<0.0001). (**D**) Venn diagram depicting unique and common differentially expressed genes between clean and *B. burgdorferi*-infected mice feeding. The up arrow indicates upregulation, and the down arrow indicates downregulation of differentially expressed genes. (**E**) Metabolism pathways inferred by GO and KEGG enrichment analyses of transcriptomes comparison between ds *GFP* and ds *ISARL* injection after feeding on *B. burgdorferi*-infected mice to repletion. (**F**) qPCR validation of *PTDSS1* showed that *PTDSS1* is positively regulated by ISARL. (**G**) Western blot of PTDSS1 protein showed that PTDSS1 is positively regulated by ISARL (**p<0.01). (**H**) qPCR assessment of *PTDSS1* transcript level, nymphal engorgement weights, and *B. burgdorferi flaB* levels in guts following RNAi silencing of *PTDSS1* after feeding on *B. burgdorferi*-infected mice. Each data point represents one nymph. (**I**) *PTDSS1* is significantly induced in the nymphal tick gut after feeding on *B. burgdorferi*-infected mice. (**J**) PTDSS1 is involved in phospholipid pathway. Cytidine diphosphate diacylglycerol (CDP-DAG) is converted to phosphatidylserine (PS) by PTDSS1. PE, phosphatidylethanolamine; PC, phosphatidylcholine. (**K**) qPCR assessment of phosphatidylserine decarboxylase (*PSD*) transcript level, nymphal engorgement weights, and qPCR assessment of *B. burgdorferi flaB* levels in guts following RNAi silencing of *PSD* after feeding on *B. burgdorferi*-infected mice. Each data point represents one nymph. Horizontal bars in the above figures represent the median. Statistical significance was assessed using a nonparametric Mann–Whitney test (ns, p>0.05; *p<0.05; **p<0.01).

The online version of this article includes the following source data and figure supplement(s) for figure 2:

**Source data 1.** Source data for PTDSS1 protein relative quantification.

**Source data 2.** Source data for PTDSS1 protein relative quantification.

**Source data 3.** PTDSS1 is regulated by ISARL and is involved in *B. burgdorferi* colonization.

**Figure supplement 1.** qPCR validation of 18 well-annotated and metabolism-related differentially expressed genes.

**Figure supplement 1—source data 1.** Source data for qPCR validation.

**Figure supplement 2.** QPCR validation of differentially expressed genes from the RNA-seq dataset.

**Figure supplement 2—source data 1.** Source data for qPCR validation.

**Figure supplement 3.** Silencing of differentially expressed genes and effects on *B. burgdorferi* acquisition.

**Figure supplement 3—source data 1.** Source data for RNAi of differentially expressed genes and effects on *B. burgdorferi* acquisition.

the phospholipid metabolic pathway in tick is critical for *B. burgdorferi*, we silenced another enzyme (ISARL-unrelated), phosphatidylserine decarboxylase (*PSD*, ISCI003338), which is an important enzyme in the synthesis of PE in both prokaryotes and eukaryotes (***Voelker, 1997***). Interestingly, we also found a significantly decreased *B. burgdorferi* burden in ds *PSD*-injected tick guts (p<0.05), and PSD and PTDSS1 elicit a similar degree of reduced *B. burgdorferi* levels (***Figure 2K***). Taken together, ISARL-mediated phospholipid metabolic pathways associated with PTDSS1 have a critical role in *B. burgdorferi* colonization.

## Mammalian adiponectin regulates tick glucose metabolism pathway but has no effect on *B. burgdorferi* colonization

We further explored how the ISARL signaling pathway is activated in ticks. Although the *I. scapularis* genome encodes an adiponectin receptor homolog, an adiponectin ligand is not present, at least in currently annotated *Ixodes* genome databases. This suggests that ticks may utilize vertebrate adiponectin to activate the adiponectin receptor during a blood meal, that tick have another ligand that stimulates the receptor, or both. Since ticks are habitually exposed to adiponectin present during a bloodmeal, we examined whether the tick adiponectin receptor could interact with incoming mammalian adiponectin during blood feeding. We injected recombinant mouse adiponectin into unfed ticks and investigated whether mammalian adiponectin could activate downstream signaling of tick adiponectin receptor by RNA-seq (***Figure 3A***). The data showed that one classic downstream gene of mammalian adiponectin signaling, tick glucose-6-phosphatase (*G6p*, ISCW017459), was significantly downregulated in the presence of mammalian adiponectin (***Figure 3A***; ***Supplementary file 1***). It has been demonstrated that in mammals the binding of adiponectin to its receptor suppresses *G6p* and phosphoenolpyruvate carboxykinase (*Pck*) expression through an AMP-activated protein kinase (AMPK)-dependent mechanism, which further inhibits glycogenolysis and gluconeogenesis (***Figure 3B***; ***Kadowaki et al., 2006***). We further searched *G6p* and *Pck* homologs in *I. scapularis* genome, and two *G6p* homologs (ISCW017459 and ISCW018612) and three *Pck* homologs (ISCW001902, ISCW000524, and ISCW001905) were identified. We designated them as *G6pc1*, *G6pc2*, *Pck1*, *Pck2*, and *Pck3*, respectively. We evaluated gene expression of all these five genes after

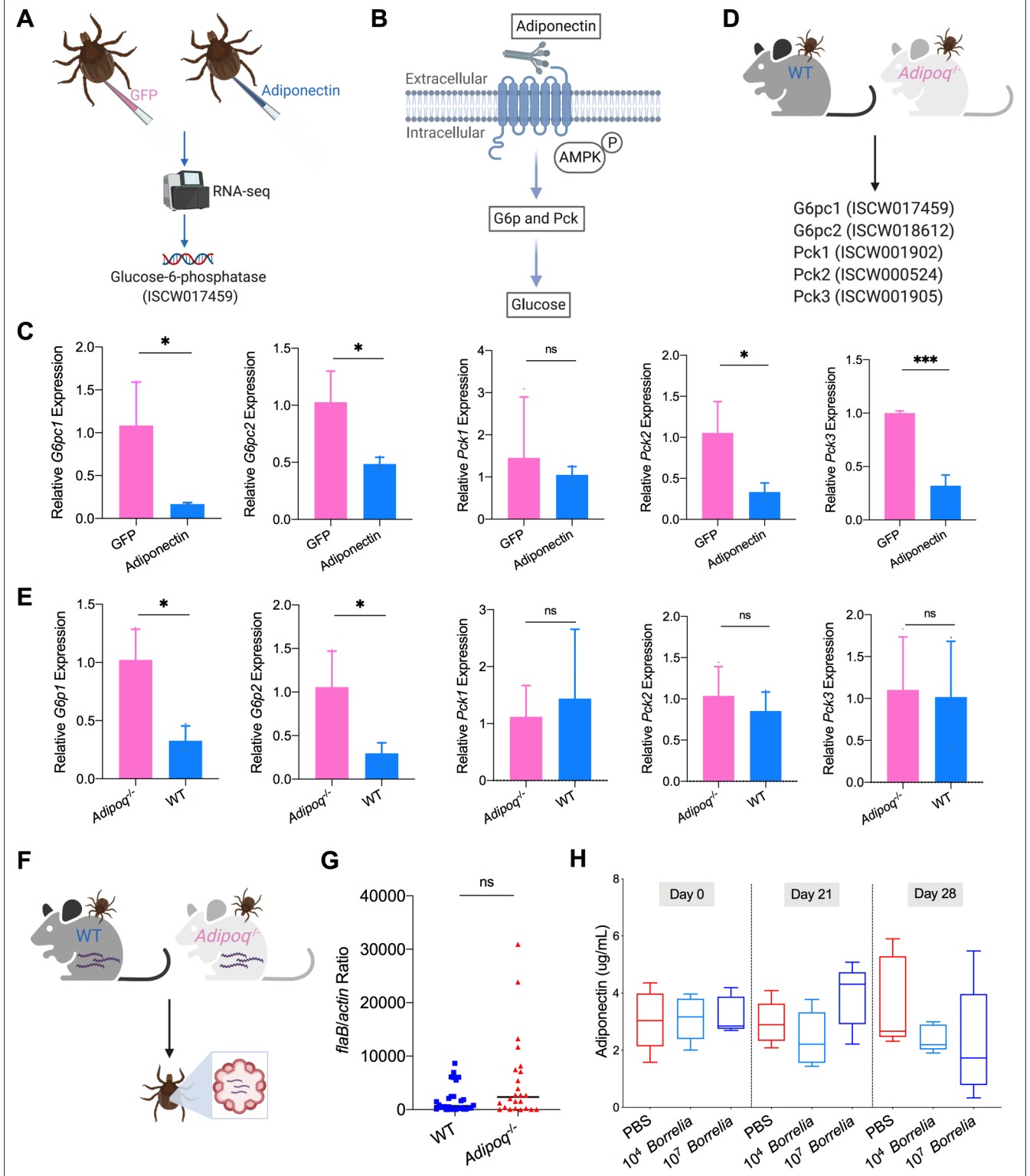

**Figure 3.** Mammalian adiponectin regulates tick glucose metabolism. (**A**) RNA-seq of injection of recombinant mouse adiponectin and GFP (control) proteins. One classic downstream gene of mammalian adiponectin receptor signaling, glucose-6-phosphatase (*G6p*), was significantly downregulated in the presence of mammalian adiponectin. (**B**) Interaction of mammal adiponectin and adiponectin receptor suppresses *G6p* and phosphoenolpyruvate carboxykinase (*Pck*) expression through an AMP-activated protein kinase (AMPK)-dependent mechanism, which further inhibits glycogenolysis and gluconeogenesis. (**C**) Injection of recombinant mouse adiponectin significantly downregulates the expression of *G6pc1*, *G6pc2*, *Pck2*, and *Pck3* in the tick gut. (**D**) Feed ticks on C57BL/6J WT and *Adipoq⁻/⁻* mice and then evaluate the expression of *G6pc1*, *G6pc2*, *Pck1*, *Pck2*, and *Pck3*. (**E**) After feeding

*Figure 3 continued on next page*

*Figure 3 continued*

on WT and *Adipoq*<sup>-/-</sup> mice, *G6pc1* and *G6pc2* showed significant downregulation profile in the presence of adiponectin, while *Pck* genes did not exhibit marked downregulation. (**F**) Ticks were fed on *B. burgdorferi*-infected WT and *Adipoq*<sup>-/-</sup> mice, and then *B. burgdorferi flaB* levels in guts were assessed. (**G**) qPCR assessment of *B. burgdorferi* burden after feeding on *B. burgdorferi*-infected WT and *Adipoq*<sup>-/-</sup> mice. No significant difference of *B. burgdorferi* burden in tick gut was observed between feeding on WT and *Adipoq*<sup>-/-</sup> mice. (**H**) Adiponectin concentration in mice sera following 21 and 28 days after injection of *B. burgdorferi* at the density of $10^4$ and $10^7$ cells/mL, respectively. Statistical significance was assessed using a nonparametric Mann–Whitney test (ns, $p > 0.05$; *$p < 0.05$; **$p < 0.01$; ***$p < 0.001$).

The online version of this article includes the following source data and figure supplement(s) for figure 3:

**Source data 1.** Mammalian adiponectin regulates tick glucose metabolism but has no effect on *B. burgdorferi* colonization.

**Figure supplement 1.** Mammalian adiponectin and tick glucose metabolism changes have no effect on *B. burgdorferi* acquisition.

**Figure supplement 1—source data 1.** Source data for effects of tick glucose metabolism on *B. burgdorferi* acquisition.

injection of recombinant adiponectin and GFP proteins. Interestingly, *G6pc1*, *G6pc2, Pck2,* and *Pck3* were significantly downregulated in the tick gut in the presence of adiponectin (*Figure 3C*). To further validate the effects on tick glucose metabolism of interaction of mammalian adiponectin and tick ISARL, we fed ticks on C57BL/6J mice deficient in adiponectin (*Adipoq*<sup>-/-</sup>) and wild-type (WT) animals, and allowed them to feed to repletion (*Figure 3D*). We then evaluated the expression of five *G6p* and *Pck* genes, and found that *G6pc1* and *G6pc2* also showed significant downregulation in the presence of adiponectin ($p < 0.05$), while *Pck* gene expression was not altered ($p > 0.05$) (*Figure 3E*).

To investigate whether the interaction of adiponectin and the receptor in ticks influences *B. burgdorferi* colonization, pathogen-free nymphs were placed on *B. burgdorferi*-infected WT and *Adipoq*<sup>-/-</sup> mice and allowed to feed to repletion (*Figure 3F*). No significant difference of the *B. burgdorferi* burden in ticks feeding on WT and *Adipoq*<sup>-/-</sup> mice was noted ($p > 0.05$) (*Figure 3G*). We also silenced the *G6pc1* and *G6pc2* genes to determine whether G6p-mediated glucose metabolic changes affect *B. burgdorferi* colonization. Consistent with the previous observation, there was no significant difference in the *B. burgdorferi* burden between control and *G6pc1*-silenced ticks ($p > 0.05$) (*Figure 3— figure supplement 1*). *G6pc2*-silenced ticks also did not show altered *B. burgdorferi* levels ($p > 0.05$) (*Figure 3—figure supplement 1*). Furthermore, the expression of *G6pc1* and *G6pc2* in the nymphs was not influenced by *B. burgdorferi* infection ($p > 0.05$) (*Figure 3—figure supplement 1*), suggesting that G6pc1- or G6pc2-mediated changes do not affect *B. burgdorferi* colonization of the tick gut. To assess any changes in the adiponectin concentration in murine serum after *B. burgdorferi* infection, the mice were injected subcutaneously with 100 μL containing $1 \times 10^4$ or $1 \times 10^7$ *B. burgdorferi,* or PBS as a control. We found that *B. burgdorferi* does not influence the adiponectin concentration in murine blood (*Figure 3H*). Taken together, these data suggest that mammalian adiponectin can regulate ISARL-mediated glucose metabolism pathway; however, it has no effect on *B. burgdorferi* colonization.

## C1QL3 is involved in the ISARL signaling pathway and modulates *B. burgdorferi* colonization

We therefore examined whether *I. scapularis* protein(s) might interact with ISARL and whether *B. burgdorferi* could influence this process – for *ISARL* silencing diminished *B. burgdorferi* colonization. To this end, we performed a blastp search of the *I. scapularis* genome with the globular C1Q domain of human and mouse adiponectin, which is known to stimulate the adiponectin receptor (*Yamauchi et al., 2002*). Two tick proteins had blastp hits with the human adiponectin C1Q domain (*Figure 4A*) and were annotated as complement C1q-like protein 3 (C1QL3) (GenBank number: XP_002415101) and conserved hypothetical protein (GenBank number: EEC18766), respectively. These are identical proteins except that C1QL3 has a signal peptide sequence, and we therefore focused on C1QL3.

We first examined whether expression of *C1QL3* could be stimulated by *B. burgdorferi* infection. A blood meal containing *B. burgdorferi* resulted in significantly increased expression of *C1QL3* in the nymphal tick guts ($p < 0.01$) (*Figure 4B*). We then generated *C1QL3*-silenced nymphs and found that these ticks had a marked reduction of the *B. burgdorferi* burden in the guts when compared to that in control *I. scapularis* ($p < 0.05$) (*Figure 4C*). This is the same observation as with silencing of *ISARL*, suggesting that *B. burgdorferi* activates the ISARL signaling pathway through the tick C1QL3 protein. Because C1QL3 C1Q domain has high similarity (64.0%) with the human adiponectin C1Q domain

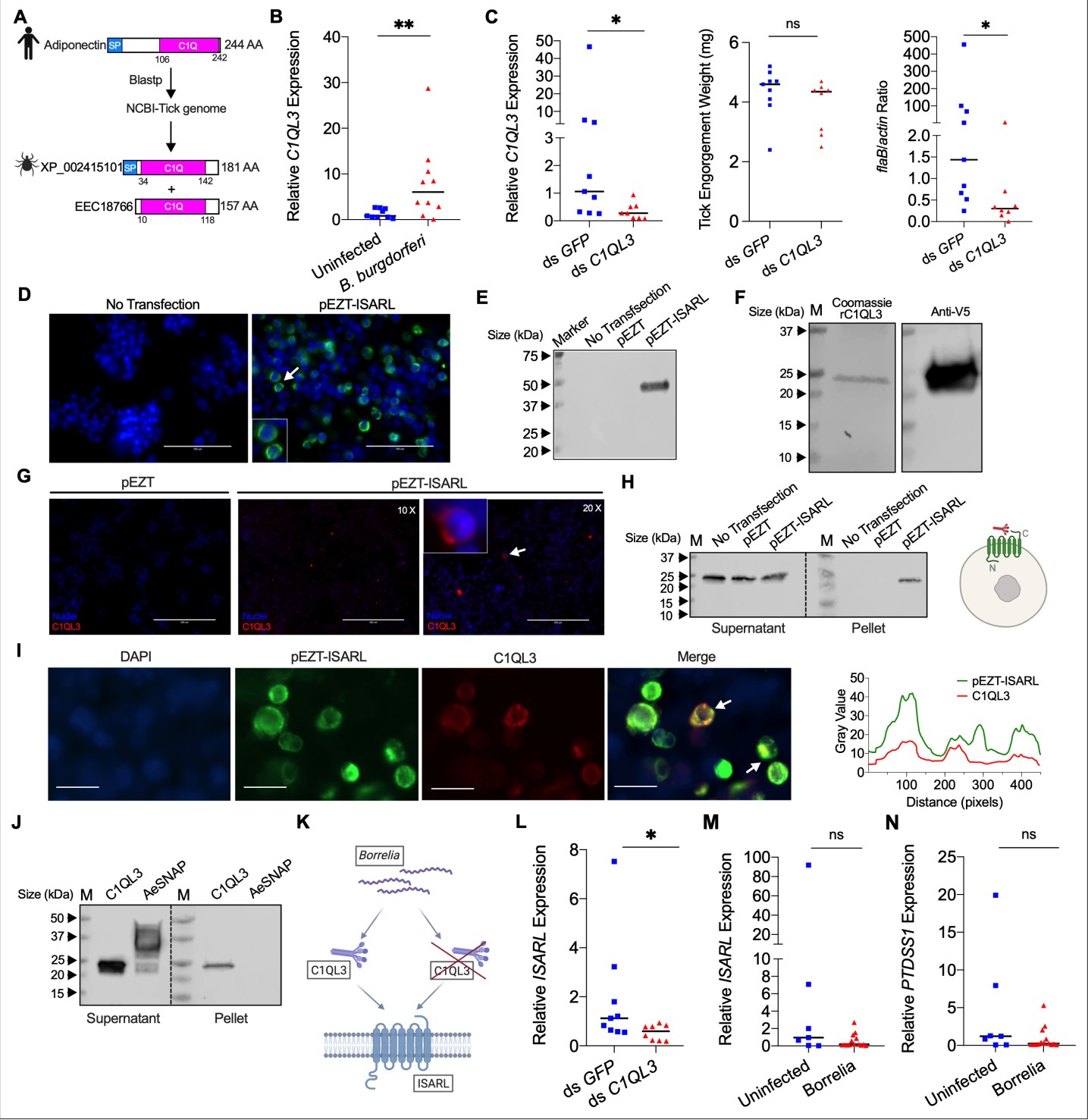

**Figure 4.** Tick complement C1q-like protein 3 (C1QL3) is involved in ISARL signaling pathways. (**A**) Blastp of the tick genome with the human adiponectin C1Q domain in NCBI generated two homologs and were annotated as complement C1q-like protein 3 (C1QL3) (GenBank number: XP_002415101) and conserved hypothetical protein (GenBank number: EEC18766), respectively. These are identical proteins except that C1QL3 has a signal peptide sequence. (**B**) *C1QL3* is significantly induced in replete nymphal tick guts after feeding on *B. burgdorferi*-infected mice. (**C**) qPCR assessment of *C1QL3* transcript levels, nymphal engorgement weights, and *B. burgdorferi* flaB levels in guts following RNAi silencing of *C1QL3* after feeding on *B. burgdorferi*-infected mice. (**D**) Human HEK293T cells were transfected with HA-tagged ISARL-expressing vector (pEZT-ISARL-HA). 40 hr post transfection, the cells were examined. The results showed that ISARL can be successfully expressed on the HEK293T cell membrane. The white arrow indicates examples of membrane expression. (**E**) Western blot confirmed ISARL expression in the HEK293T cells. (**F**) Generation of tick C1QL3

*Figure 4 continued on next page*

*Figure 4 continued*

protein with His/V5-tag in a *Drosophila* expression system. Recombinant protein was assessed by SDS-PAGE gel and western blot. (**G**) C1QL3 is bound on the membrane of ISARL-expressed HEK293T cells. 10× and 20× are the microscope magnifications. The white arrow indicates one example of binding. (**H**) Binding of C1QL3 to ISARL as analyzed by a pull-down assay. HRP V5-tag monoclonal antibody was used to detect protein. C1QL3 was only detected in ISARL-expressed cells pellet. (**I**) Co-immunolocalization of ISARL (green) and C1QL3 (red). The specific signal of C1QL3 protein was observed on the surface of some of ISARL-expressed cells, and no signal was shown on nonsuccessfully expressed cells membrane. The white arrows indicate examples of binding. Bar: 20 μm. The plot profile of co-localization was conducted by ImageJ software. (**J**) Binding of C1QL3 to tick ISE6 cells as analyzed by a pull-down assay. The *Aedes aegypti* synaptosomal-associated protein (AeSNAP) was used as control. HRP V5-tag monoclonal antibody was used to detect protein. (**K**) Analysis of how silencing of *C1QL3* influences *ISARL* expression after feeding on *B. burgdorferi*-infected mice. (**L**) qPCR assessment showed that *ISARL* transcript levels following RNAi silencing of *C1QL3* were significantly lower than in control ds *GFP*-injected tick guts after feeding on *B. burgdorferi*-infected mice. (**M**) qPCR assessment showed that a blood meal containing *B. burgdorferi* did not significantly increase expression of *ISARL* in the nymphal tick guts as compared to feeding on clean mice after RNAi silencing of *C1QL3*. (**N**) qPCR assessment showed that a blood meal containing *B. burgdorferi* did not significantly increase expression of *PTDSS1* in the nymphal tick guts as compared to feeding on clean mice after RNAi silencing of *C1QL3*. Each data point represents one nymph. Horizontal bars in the above figures represent the median. Statistical significance was assessed using a nonparametric Mann–Whitney test (ns, p>0.05; *p<0.05; **p<0.01).

The online version of this article includes the following source data and figure supplement(s) for figure 4:

**Source data 1.** Source data for ISARL expression.

**Source data 2.** Source data for C1QL3 protein purification.

**Source data 3.** Source data for C1QL3 protein purification.

**Source data 4.** Source data for binding of C1QL3 to ISARL.

**Source data 5.** Source data for binding of C1QL3 to tick ISE6 cells.

**Source data 6.** C1QL3 is involved in the ISARL signaling pathway and modulates *B. burgdorferi* colonization.

**Figure supplement 1.** Alignment of human adiponectin C1Q domain and C1QL3 C1Q domain.

(*Figure 4—figure supplement 1*), and C1Q proteins have been demonstrated to activate diverse pathways through the adiponectin receptor (*Zheng et al., 2011*), we investigated whether tick C1QL3 could interact with ISARL. Human embryonic kidney HEK293T cells were transfected with the ISARL expression vector (pEZT-ISARL). The results showed that tick ISARL can be successfully expressed, as validated by cell staining and western blot (*Figure 4D and E*), on the HEK293T cell membrane (*Figure 4D*). We then generated tick C1QL3 protein in a *Drosophila* expression system (*Figure 4F*). The HEK293T cells were then incubated with the recombinant C1QL3 protein. After washing and staining, recombinant C1QL3 could be detected on the surface of ISARL-expressed rather than empty plasmid-transfected HEK293T cells (*Figure 4G*). A pull-down assay also indicated that recombinant C1QL3 interacts with ISARL as demonstrated by the detection of C1QL3 only in ISARL-expressed cells pellet (*Figure 4H*). In addition, co-immunolocalization demonstrated that the C1QL3 protein specifically binds to the ISARL-expressed cell membrane (*Figure 4I*). Furthermore, C1QL3 also bound to tick ISE6 cells, a non-heterologous system (*Figure 4J*).

Since C1QL3 is a ligand of tick ISARL and also involved in *Borrelia* colonization, we further investigated whether C1QL3 has a role on the activation of ISARL by *Borrelia*. We first assessed if silencing of *C1QL3* influenced *ISARL* expression after feeding on *B. burgdorferi*-infected mice (*Figure 4K*). qPCR assessment showed that the *ISARL* transcript level following RNAi silencing of *C1QL3* was significantly lower than that in control ds *GFP*-injected tick guts after feeding on *B. burgdorferi*-infected mice (p<0.05) (*Figure 4L*). More importantly, after silencing *C1QL3*, a blood meal containing *B. burgdorferi* did not significantly increase expression of *ISARL* and *PTDSS1* in the nymphal tick guts as compared to feeding on clean mice (p>0.05) (*Figure 4M and N*), further suggesting that C1QL3 plays a role in the ISARL signaling and phospholipid metabolism pathways.

## Discussion

Adiponectin is a hormone, secreted mainly from adipocytes, that stimulates glucose utilization and fatty acid oxidation (*Berg et al., 2001*; *Fruebis et al., 2001*). The key roles of adiponectin in regulating energy homeostasis are mediated by adiponectin receptors across species including humans, yeast, nematodes, and flies (*Kwak et al., 2013*; *Narasimhan et al., 2005*; *Svensson et al., 2011*; *Yamauchi et al., 2003*). In this study, we have identified and characterized an adiponectin receptor homologue from *I. scapularis*, ISARL. ISARL shares significant sequence similarities with human,

mouse, and *Drosophila* adiponectin receptors. In addition, ISARL contains the canonical features of adiponectin receptors, including conserved TM domains, a long internal N-terminal region, and a relatively short external C-terminal region. The highly conserved amino acids and the structures of ISARL and the receptor from *D. melanogaster* suggest that their ligands and signaling pathways may also be conserved in arthropods. However, homologs of adiponectin have not yet been identified in arthropods, suggesting that ligands for adiponectin receptors in arthropods may interact in different ways than in vertebrates.

The Lyme disease agent, *B. burgdorferi*, engages in intimate interactions with *I. scapularis* during its acquisition and colonization of the tick gut (*Radolf et al., 2012*). This is accompanied by dramatic changes in the expression profiles of *Borrelia* and tick gut genes, which are critical drivers for colonization, persistence, or transmission (*Kurokawa et al., 2020*; *Narasimhan et al., 2017*). In our study, expression of *ISARL* was significantly increased in the nymphal tick gut after *B. burgdorferi* infection. The upregulation of *ISARL* correlates with *Borrelia* infection in the gut. More interestingly, after silencing *ISARL* expression in the tick gut by anal pore injection, nymphal tick guts showed a marked reduction in the *B. burgdorferi* burden when compared to that in control ticks, demonstrating that ISARL facilitates *B. burgdorferi* colonization.

We utilized RNA-seq to elucidate the pathways that are altered when *ISARL* is silenced in ticks that engorge on clean and *B. burgdorferi*-infected mice. Of note, ISARL can regulate a critical enzyme involved in phospholipid metabolism, PTDSS1. Regulation of PTDSS1 by adiponectin receptors is also found in other organisms such as yeast, where the adiponectin receptor homolog Izh2 is connected to phospholipid metabolism through co-regulation of the expression of inositol-3-phosphate synthase (*INO1*) and phosphatidylserine synthase (*CHO1*, homolog of *PTDSS1*) genes with zinc-responsive activator protein (*Zap1*) (*Mattiazzi Ušaj et al., 2015*). Silencing of *I. scapularis* *PTDSS1* led to a reduced spirochete burden in the guts, thereby linking *B. burgdorferi* colonization with phospholipid metabolism. The *B. burgdorferi* genome is small and encodes a limited number of metabolic enzymes (*Casjens et al., 2000*; *Fraser et al., 1997*). The restricted biosynthetic capability forces *B. burgdorferi* to rely on its vertebrate and arthropod hosts for nutrients or enzymes that it cannot synthesize (*Tilly et al., 2008*). Interestingly, we also found that silencing of *I. scapularis* *3HADH*, which is involved in fatty acid metabolic processes, decreased the *B. burgdorferi* burden in tick gut (*Figure 2—figure supplement 3*). The markedly decreased *B. burgdorferi* burden in ticks after silencing of *PTDSS1*, *PSD*, and *3HADH* suggests that the spirochete may require the tick for selected metabolic needs. Indeed, previous studies have demonstrated that feeding ticks provide Lyme disease spirochetes with glycerol, an alternative carbohydrate energy source and essential building block for phospholipid biosynthesis (*Kerstholt et al., 2020*; *Pappas et al., 2011*). In addition, *B. burgdorferi* can also acquire lipids from the membranes of eukaryotic cells to which they are attached (*Crowley et al., 2013*).

We also found that *B. burgdorferi* can upregulate an adiponectin-related protein, C1QL3, in ticks, which associates with ISARL and leads to phospholipid metabolism changes in ticks. We propose that C1QL3 in tick is mainly involved in metabolism, rather than complement activation, as demonstrated by the decreased *B. burgdorferi* level after silencing C1QL3. Indeed, some of C1Q/TNF family proteins are associated with metabolism. In addition to adiponectin, proteins such as C1Q/TNF-related protein 3 (CTRP3), CTRP5, CTRP9, CTRP13 (C1QL3), and CTRP15 also belong to adipokine family and have been reported to be associated with the regulation of glucose, lipid, or other metabolisms (*Jiang et al., 2018*; *Li et al., 2017*; *Mi et al., 2019*; *Wei et al., 2011*; *Wolf et al., 2016*). Importantly, C1Q proteins have been demonstrated to activate diverse pathways through adiponectin receptor (*Zheng et al., 2011*). Additional efforts will investigate the mechanisms by which *B. burgdorferi* influence C1QL3 expression, and whether C1QL3 homologs in mammals such as adiponectin, CTRP13, or other C1Q/TNF-related proteins may stimulate the tick C1QL3/ISARL pathway.

Adiponectin receptors have diverse essential functions, and mutations in adiponectin receptors result in critical deficiencies. For instance, mutant of the AdipoR1 gene in retinal pigment epithelial cells results in the inability to take up and retain the essential fatty acid family member docosahexaenoic acid (DHA, 22:6,n-3), further leading to photoreceptor cell death and retinal degeneration (*Rice et al., 2015*; *Sluch et al., 2018*). Adiponectin receptors are thought to have ceramidase activity (*Vasiliauskaité-Brooks et al., 2017*), which is critical for cell survival through formation of antiapoptotic metabolite-sphingosine-1-phosphate (S1P) (*Holland et al., 2011*). Whether targeting the tick

adiponectin receptor signaling or the adiponectin pathway has the ability to influence human infection with *B. burgdorferi* remains to be determined.

In summary, we demonstrate that ISARL plays a key role in metabolic pathways in *I. scapularis*. ISARL-mediated phospholipid metabolism by PTDSS1 contributes to *B. burgdorferi* colonization and an adiponectin-related protein, C1QL3, is involved in ISARL signaling pathway. These studies elucidate a new pathway involved in tick metabolism and demonstrate that *B. burgdorferi* co-opts the activation of this pathway to facilitate colonization of *I. scapularis*. These processes are crucial to understanding the complex life cycle of the Lyme disease agent within ticks, and may be applicable to other arthropods and arthropod-borne infectious agents.

# Materials and methods

## Key resources table

| Reagent type (species) or resource | Designation | Source or reference | Identifiers | Additional information |
|---|---|---|---|---|
| Biological sample (*Mus musculus*) | C3H/HeJ | Jackson Laboratory | Stock #: 000659; RRID:IMSR_JAX:000659 | |
| Biological sample (*M. musculus*) | WT C57BL/6J | Jackson Laboratory | Stock #: 000664; RRID:IMSR_JAX:000664 | |
| Biological sample (*M. musculus*) | Adipoq$^{-/-}$ C57BL/6J | Jackson Laboratory | Stock #: 008195; RRID:IMSR_JAX:008195 | |
| Biological sample (*Borrelia burgdorferi*) | *Borrelia burgdorferi* strain N40 | Dr. Erol Fikrig Laboratory | | |
| Biological sample (*Ixodes scapularis*) | Black-legged tick | Oklahoma State University | | |
| Cell line (*Homo sapiens*) | Human embryonic kidney HEK293T | ATCC | #CRL-3216; RRID:CVCL_0063 | |
| Cell line (*I. scapularis*) | Tick ISE6 | ATCC | #CRL-11974; RRID:CVCL_Z170 | |
| Antibody | Anti-HA (rabbit monoclonal) | Cell Signaling Technology | #C29F4; RRID:AB_10693385 | IF (1:100) |
| Antibody | Anti-V5 (mouse monoclonal) | Invitrogen | #R960-25; RRID:AB_2556564 | IF (1:100) |
| Antibody | Goat anti-rabbit IgG (H + L) Highly Cross-Adsorbed Secondary Antibody, Alexa Fluor 488 | Invitrogen | #A-11034; RRID:AB_2576217 | IF (1:100) |
| Antibody | Goat anti-mouse IgG (H + L) Cross-Adsorbed Secondary Antibody, Alexa Fluor 555 | Invitrogen | #A-21422; RRID:AB_141822 | IF (1:100) |
| Antibody | HRP Anti-His tag antibody (chicken polyclonal) | Abcam | #ab3553; RRID:AB_303900 | WB (1:10,000) |
| Antibody | HRP V5-tag (mouse monoclonal) | Invitrogen | #R961-25; RRID:AB_2556565 | WB (1:1000) |
| Peptide, recombinant protein | Mouse adiponectin | MilliporeSigma | #SRP3297 | |
| Peptide, recombinant protein | *Aequorea victoria* green fluorescent protein (GFP) | SinoBiological | #13105-S07E | |
| Commercial assay or kit | Mouse adiponectin/Acrp30 Quantikine ELISA Kit | R&D Systems | #MRP300; RRID:AB_2832917 | |
| Software, algorithm | Prism | GraphPad | RRID:SCR_002798 | |

## Mice, spirochetes, ticks, and cell lines

C3H/HeJ mice, C57BL/6J mice WT, and C57BL/6J mice deficient in adiponectin (*Adipoq$^{-/-}$*) were purchased from the Jackson Laboratory (https://www.jax.org/strain/008195). All mice were bred and maintained in a pathogen-free facility at Yale University. The spirochetes *B. burgdorferi* N40 were grown at 33°C in Barbour–Stoenner–Kelly H (BSK-H) complete medium (Sigma-Aldrich, #B8291) with 6% rabbit serum. The live cell density was ~$10^6$–$10^7$ cells/mL as determined by dark field microscopy and hemocytometric analysis. To obtain *B. burgdorferi*-infected mice, the mice were injected

subcutaneously with 100 μL of *B. burgdorferi* N40 ($1 \times 10^5$ cells/mL). Two weeks after inoculation, the *B. burgdorferi* burden in mice was assayed by qPCR analysis of spirochete DNA in murine ear punch biopsies as described below. DNA was extracted from mouse skin-punch biopsies using the DNeasy tissue kit (QIAGEN, #69506) according to the manufacturer's protocol. The DNA was analyzed by qPCR using *flagellinB* (*flaB*) primers, and data were normalized to mouse *actin*. The primer sequences are shown in *Supplementary file 1*. Pathogen-free *I. scapularis* larvae were acquired from the Centers for Disease Control and Prevention. The larval ticks were fed to repletion on pathogen-free C3H/HeJ mice and allowed to molt to nymphs. *B. burgdorferi*-infected nymphs were generated by placing larvae on *B. burgdorferi*-infected C3H/HeJ mice, and fed larvae were molted to nymphs. Nymphal ticks were maintained at 85% relative humidity with a 14 hr light and 10 hr dark period at 23°C. Human embryonic kidney HEK293T cells (ATCC, #CRL-3216) and tick ISE6 cells (ATCC, #CRL-11974) were used for vitro studies. The identity of the cells has been authenticated by ATCC, and no mycoplasma contamination.

## Identification and characterization of the *I. scapularis* adiponectin receptor-like (*ISARL*) gene

The human adiponectin receptor protein 1 (GenBank number: NP_001277482) and 2 (GenBank number: NP_001362293) sequences were used to conduct tblastn and blastp searches against the available black-legged tick database (taxid:6945) using NCBI default parameters. Tick adiponectin receptor sequence was further validated by amplification with primers in *Supplementary file 1*. Multiple alignment of protein sequences were performed using the Clustal Omega (https://www.ebi.ac.uk/Tools/msa/clustalo/; *Madeira et al., 2019*) or Uniprot (https://www.uniprot.org/align/). The similarities of adiponectin receptor protein sequences were measured in EMBOSS supermatcher (https://www.bioinformatics.nl/cgi-bin/emboss/supermatcher). The protein structure of ISARL was predicted in SWISS-MODEL (https://swissmodel.expasy.org/ *Guex and Peitsch, 1997*; *Waterhouse et al., 2018*). Hydrophobicity analysis was performed using ProtScale (https://web.expasy.org/protscale/; *Gasteiger et al., 2005*).

## Tick exposure to *B. burgdorferi* and expression of *ISARL*

To evaluate gene expression of *ISARL* upon *B. burgdorferi* infection, pathogen-free *I. scapularis* nymphs were placed on *B. burgdorferi*-free and -infected mice (C3H/HeJ). At least three mice were used in each experiment, and the ticks were allowed to feed to repletion. Both *B. burgdorferi*-free and -exposed tick guts were dissected under the dissecting microscope. The RNA from dissected guts was purified by TRIzol (Invitrogen, #15596-018) following the manufacturer's protocol, and cDNA was synthesized using the iScript cDNA Synthesis Kits (Bio-Rad, #1708891). qPCR was performed using iQ SYBR Green Supermix (Bio-Rad, #1725124) on a Bio-Rad cycler with a program consisting of an initial denaturing step of 2 min at 95°C and 45 amplification cycles consisting of 20 s at 95°C followed by 15 s at 60°C, and 30 s at 72°C. The genes and corresponding primer sequences are shown in *Supplementary file 1*. The specific target transcripts of *ISARL* and the reference gene tick *actin* were quantified by extrapolation from a standard curve derived from a series of known DNA dilutions of each target gene, and data were normalized to tick *actin*.

## RNAi silencing of targeted genes

Fed-nymph gut cDNA was prepared as described above and used as template to amplify segments of targeted genes. The PCR primers with T7 promoter sequences are shown in *Supplementary file 1*. Double-stranded RNA (dsRNA) were synthesized using the MEGAscript RNAi kit (Invitrogen, #AM1626M) using PCR-generated DNA template that contained the T7 promoter sequence at both ends. The dsRNA quality was examined by agarose gel electrophoresis. DsRNA of the *Aequorea victoria* green fluorescent protein (GFP) was used as a control. Pathogen-free and -infected tick nymphs were injected in the anal pore with dsRNA (6 nL) using glass capillary needles as described by *Narasimhan et al., 2004*.

## Effects of silenced genes on *B. burgdorferi* colonization and transmission

To examine the effects of silencing targeted genes on the colonization of *B. burgdorferi* in the tick gut, dsRNA microinjected pathogen-free *I. scapularis* nymphs were placed on *B. burgdorferi*-infected mice (C3H/HeJ) and allowed to feed to repletion. The ticks were then collected for gut dissection. The *B. burgdorferi* burden in the tick gut was quantified by amplifying *flaB*. FlaB was quantified by extrapolation from a standard curve derived from a series of known DNA dilutions of *flaB* gene, and data were normalized to tick *actin*. The knockdown efficiency of targeted genes was tested as described above. Specifically, the expression of targeted genes was estimated with the $\Delta\Delta C_T$ method (*Schmittgen and Livak, 2008*) using the reference gene *actin*. To test the effects of silencing *ISARL* on the transmission of *B. burgdorferi*, a group of 3–5 *GFP* or *ISARL* dsRNA-injected *B. burgdorferi*-infected nymphs were placed on each C3H/HeJ mouse (at least five mice each in the *GFP* or *ISARL* dsRNA groups) and allowed to feed to repletion. Ticks are placed on the mouse head/back between the ears. At 7 and 14 days post tick detachment, the mice were anesthetized, and skin was aseptically punch biopsied and assessed for spirochete burden by qPCR. Ticks feed in head area and skin punch biopsies are collected from the pinnae/ears. This site is considered distal as it is not at the site of tick bite. Twenty-one days post tick detachment, the mice were sacrificed, and ear skin, heart, and joints were aseptically collected and assessed for spirochete burden by qPCR.

## RNA-seq and bioinformatic analyses

dsRNA (ds *ISARL* and ds *GFP*) microinjected pathogen-free *I. scapularis* nymphs were placed on clean and *B. burgdorferi*-infected mice (C3H/HeJ), respectively, and allowed to feed to repletion. Then, the ticks were collected for gut dissection. Total RNA was purified as described above. In addition, to check the transcriptional alterations in the tick gut in the presence of mammalian adiponectin, pathogen-free tick nymphs were injected in the anal pore with approximately 12 ng recombinant mouse adiponectin (MilliporeSigma, #SRP3297) and GFP proteins (SinoBiological, #13105-S07E). The amount of injected protein was calculated based on the adiponectin concentration in mice blood (~3 µg/mL) and nymphal tick engorgement (~4 mg). Then, the guts were dissected after 8 hr injection, and RNA was purified. The RNA samples were then submitted for library preparation using TruSeq (Illumina, San Diego, CA) and sequenced using Illumina HiSeq 2500 by paired-end sequencing at the Yale Centre for Genome Analysis (YCGA). The *I. scapularis* transcript data were downloaded from the VectorBase (https://vectorbase.org/vectorbase/app/ *Giraldo-Calderón et al., 2015*) and indexed using the kallisto-index (*Bray et al., 2016*). The reads from the sequencer were pseudo-aligned with the index reference transcriptome using kallisto (*Bray et al., 2016*). The counts generated from three biological replicates each treatment were processed by DESeq2 (*Love et al., 2014*) in RStudio (https://rstudio.com). The significant genes were then determined by the p-value and the adjusted p-value padj (p<0.05). The heatmaps of significant genes were also generated in RStudio. GO enrichment analysis and KEGG pathway enrichment analyses were conducted using the functional annotation tool DAVID 6.8 (*Huang et al., 2009*).

## Expression of ISARL and binding assays

Tick *ISARL* gene was PCR amplified from nymph cDNA using the primer pair listed in *Supplementary file 1*, then cloned into the *Xba*I and *Not*I sites of the pEZT-Dlux, a modified pEZT-BM vector (Addgene, #74099) in-frame with a HA-tag sequence, by Gibson Assembly Cloning Kit (NEB, #E5510S). The HEK293T cells were grown in Dulbecco's Modified Eagle's Medium (DMEM, Thermo Fisher, #11965-118) media supplemented with 10% fetal bovine serum (FBS, Sigma, #12306C-500). HEK293T cells were transfected with the *ISARL* expression plasmid (pEZT-ISARL) using TransIT 2020 (Mirus, #MIR5404). After 40 hr post transfection, the cells were washed with 1× PBS and then incubated with 5 µg rC1QL3 protein (0.5 µg/µL) with His/V5 tag, respectively. After 16 hr incubation with gentle agitation, the cells were washed with PBS and fixed in 4% PFA for 15 min at room temperature. Then, the cells were blocked in 1% BSA in PBS for 1 hr and subsequently immunolabeled with anti-HA antibody (1:100, Cell Signaling Technology, #C29F4) for checking ISARL expression and V5 tag monoclonal antibody (1:100, Invitrogen, # R960-25) for checking C1QL3 binding. Cells were washed with PBS three times and then immunolabeled with secondary antibodies of goat anti-rabbit IgG (H + L) Highly Cross-Adsorbed Secondary Antibody, Alexa Fluor 488 (1:100, Invitrogen,

#A-11034) and goat anti-mouse IgG (H + L) Cross-Adsorbed Secondary Antibody, Alexa Fluor 555 (1:100, Invitrogen, #A-21422) for 1 hr at room temperature. Nuclei were stained with DAPI (Invitrogen, #D9542). After staining, the fluorescence signals were examined with an EVOS FL Auto Cell Imaging System (Thermo Fisher Scientific). We also conducted plot profile to help analyze co-localization by ImageJ software.

For checking ISARL expression by western blot, after 40 hr post transfection, the cells were washed with 1× PBS and then lysed with 4× Laemmli Sample Buffer (Bio-Rad, #1610747). After centrifuging at high speed, the supernatant was loaded to perform western blot as described below. HRP anti-His tag antibody (1:10,000, Abcam, #ab3553) or anti-HA antibody (1:1000, Cell Signaling Technology, #C29F4) was used to detect expression of ISARL.

We conducted a pull-down assay to check the binding of ISARL and C1QL3 as described in *Schuijt et al., 2011a*. Briefly, HEK293T cells were transfected as described above. After 40 hr post transfection, the cells were washed and suspended with 1× PBS and then incubated with rC1QL3 protein for 16 hr with gentle agitation, respectively. Then the cells were pelleted, and the pellet and supernatant were separated. The pellet was washed 5–8 times in 1.5 mL PBS/0.1% BSA and was resuspended in the same volume as the supernatant. The test of C1QL3 binding to tick ISE6 cells was conducted as described above. Equal volumes of supernatant and pellet were used to run western blot as described below. HRP V5-tag monoclonal antibody (1:1000, Invitrogen, # R961-25) was used to detect protein.

## Adiponectin concentration in serum after *B. burgdorferi* infection

To assess the adiponectin concentration change in mice serum after *B. burgdorferi* infection, the C3H/HeJ mice were injected subcutaneously with 100 μL $1 \times 10^4$ and $1 \times 10^7$ cells/mL *B. burgdorferi* and PBS as a control (five mice in each group). At 0, 21, and 28 days post inoculation, the blood was collected from mice. The sera were separated from mice blood samples by centrifugation at 1000× g for 10 min at 4°C. The adiponectin in mice serum was quantified by Mouse Adiponectin/Acrp30 Quantikine ELISA Kit (R&D Systems, #MRP300).

## Effects of adiponectin in mice blood on *B. burgdorferi* colonization

Pathogen-free *I. scapularis* nymphs were placed on *B. burgdorferi*-infected WT and *Adipoq*[-/-] mice (C57BL/6J) and allowed to feed to repletion. The ticks were then collected for gut dissection. The *B. burgdorferi* burden in the tick gut was quantified as described above.

## Purification of recombinant proteins

The *C1QL3* was PCR amplified from tick nymph cDNA using the primer pair listed in *Supplementary file 1*, then cloned into the *Bgl*II and *Xho*I sites of the pMT/BiP/V5-His vector (Invitrogen, #V413020). The recombinant protein was expressed and purified using the *Drosophila* Expression System as described previously (*Schuijt et al., 2011b*). The protein was purified from the supernatant by TALON metal affinity resin (Clontech, #635606) and eluted with 150 mM imidazole. The eluted samples were filtered through a 0.22 μm filter and concentrated with a 10 kDa concentrator (MilliporeSigma, #Z740203) by centrifugation at 4°C. Recombinant protein purities were assessed by SDS-PAGE using 4–20% Mini-Protean TGX gels (Bio-Rad, #4561094) and quantified using the BCA Protein Estimation kit (Thermo Fisher Scientific, #23225).

## Western blot

Proteins were separated by SDS-PAGE at 160 V for 1 hr. Proteins were transferred onto a 0.45-m-pore-size polyvinylidene difluoride (PVDF) membrane (Bio-Rad, #1620177) and processed for immunoblotting. The blots were blocked in 1% non-fat milk in PBS for 60 min. Primary antibodies of PTDSS1 rabbit pAb (1:1000, Abclonal, #A13065), anti-beta actin antibody (1:1000, Abcam, #ab8224), HRP anti-6X His tag antibody (1:10,000, Abcam, #ab3553), or HRP V5 tag monoclonal antibody (1:1000, Invitrogen, # R961-25) were diluted in 0.05% PBST and incubated with the blots for 1 hr at room temperature or 4°C overnight. HRP-conjugated secondary antibody (1:2500, Invitrogen, #62-6520 and #31466) was diluted in PBST and incubated for 1 hr at room temperature. After washing with PBST, the immunoblots were imaged and quantified with a LI-COR Odyssey imaging system.

## Statistical analysis

Statistical significance of differences observed in experimental and control groups was analyzed using GraphPad Prism version 8.0 (GraphPad Software, Inc, San Diego, CA). Nonparametric Mann–Whitney test or unpaired t test were utilized to compare the mean values of control and tested groups, and $p<0.05$ was considered significant. The exact p-values are shown in the source data.

## Acknowledgements

This work was supported by grants from the NIH (AI126033, AI138949) and the Steven and Alexandra Cohen Foundation. We sincerely thank Ms Kathleen DePonte for her excellent technical assistance. We would like to acknowledge that figures were created using BioRender.

## Additional information

### Funding

| Funder | Grant reference number | Author |
| --- | --- | --- |
| National Institutes of Health | AI126033 | Erol Fikrig |
| National Institutes of Health | AI138949 | Erol Fikrig |
| Steven and Alexandra Cohen Foundation | | Erol Fikrig |

The funders had no role in study design, data collection and interpretation, or the decision to submit the work for publication.

### Author contributions

Xiaotian Tang, Conceptualization, Data curation, Formal analysis, Methodology, Software, Validation, Visualization, Writing – original draft, Writing – review and editing; Yongguo Cao, Conceptualization, Data curation, Formal analysis, Methodology, Validation, Visualization, Writing – review and editing; Gunjan Arora, Data curation, Formal analysis, Software, Writing – review and editing; Jesse Hwang, Conceptualization, Data curation, Investigation, Methodology, Writing – review and editing; Andaleeb Sajid, Courtney L Brown, Alejandro Marín-López, Hongwei Ma, Data curation, Methodology, Writing – review and editing; Sameet Mehta, Data curation, Methodology, Software, Writing – review and editing; Yu-Min Chuang, Data curation, Investigation, Writing – review and editing; Ming-Jie Wu, Data curation, Investigation, Methodology, Writing – review and editing; Utpal Pal, Conceptualization, Project administration, Resources, Supervision, Writing – review and editing; Sukanya Narasimhan, Conceptualization, Supervision, Writing – review and editing; Erol Fikrig, Conceptualization, Funding acquisition, Project administration, Resources, Supervision, Writing – original draft, Writing – review and editing

### Author ORCIDs

Xiaotian Tang  http://orcid.org/0000-0002-0171-9354
Yongguo Cao  http://orcid.org/0000-0002-9533-7516
Courtney L Brown  http://orcid.org/0000-0001-7423-3331
Yu-Min Chuang  http://orcid.org/0000-0003-2241-5541
Erol Fikrig  http://orcid.org/0000-0002-5884-6047

### Ethics

Animal care and housing were performed according to the Guide for the Care and Use of laboratory Animals of National Institutes of Health, USA. All protocols in this study were approved by the Yale University Institutional Animal Care and Use Committee (YUIACUC) (approval number 2018-07941).

### Decision letter and Author response

Decision letter https://doi.org/10.7554/eLife.72568.sa1
Author response https://doi.org/10.7554/eLife.72568.sa2

## Additional files

### Supplementary files

• Transparent reporting form

• Supplementary file 1. Supplementary files in this study. (a) Summary of differently expressed genes of comparison between ds GFP and ds ISARL injection after 96 hr feeding on clean mice. (b) Summary of differently expressed genes of comparison between ds GFP and ds ISARL injection after 96 hr feeding on *B. burgdorferi*-infected mice. (c) Summary of differently expressed genes of comparison between recombinant GFP and adiponectin proteins injection after 8 hr. (d) The primers used in this study.

### Data availability

The RNA-seq data are available in the Gene Expression Omnibus (GEO) repository at the National Center for Biotechnology Information under the accession number: GSE169293.

The following dataset was generated:

| Author(s) | Year | Dataset title | Dataset URL | Database and Identifier |
|---|---|---|---|---|
| Tang X | 2021 | The Lyme Disease agent co-opts adiponectin receptor-mediated signaling in its arthropod vector | https://www.ncbi.nlm.nih.gov/geo/query/acc.cgi?acc=GSE169293 | NCBI Gene Expression Omnibus, GSE169293 |

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
