## [Editor Report]

This work is a superb demonstration of how *B. burgdorferi* hijacks the ISARL-mediated phospholipid metabolism pathway to facilitate survival inside *Ixodes scapularis* ticks. Unexpectedly, the authors show that *B. burgdorferi* upregulates the tick complement C1q-like protein 3, which interacts with ISARL, to provide the required metabolic needs for the bacterium.

---

## [Decision Letter]

**Decision letter after peer review:**

Thank you for submitting your article "The Lyme Disease agent co-opts adiponectin receptor-mediated signaling in its arthropod vector" for consideration by *eLife*. Your article has been reviewed by 3 peer reviewers, and the evaluation has been overseen by Shaeri Mukherjee (Reviewing Editor) and Philip Cole (Senior Editor). The following individual involved in the review of your submission has agreed to reveal their identity: Brandon Garcia (Reviewer #3).

Essential revisions:

Your manuscript was very well-received by the three reviewers and we are happy to consider a revised manuscript for publication at e*Life*. Given that most of the comments focus on clarity and not so much on new data, it is essential that in your revision, you provide: (1) An improved Discussion section (2) Clarity on methods and references described (3) Provide explanation on the significance of the data. All three reviewers' comments need to be addressed before the manuscript can be accepted.

*Reviewer #1 (Recommendations for the authors):*

This is a technically impressive study that clearly breaks new ground in our understanding of the influence of Lyme disease spirochetes on the transcriptional/metabolic responses of tick midguts during an acquisition blood meal. The authors should address the following concerns:

1. Textual clarity/interpretation of data:

a. The ISARL knockdown RNAseq experiments (lines 163-201). The description of the results should be revised to better reflect the information in the corresponding tables (what is meant by "amino acid"). The two paragraphs on page 9 (lines 171-192) are particularly difficult to follow. One wonders what to make of these data given the fact that so many of the differentially regulated genes (13 of 17) identified by RNAseq were not confirmed by qRT-PCR (Figure S3). Also see comments on Figure 2 below.

b. It appears that mouse adiponectin injection experiments (p12) were done with flat ticks. Since midguts in flat ticks are metabolically quiescent, the rationale for this isn't clear. It is surprising that adiponectin effects in flat and engorged midguts are similar.

c. Lines 399-401 in the discussion. The authors do not demonstrate that *B. burgdorferi* interacts with C1QL3, nor do they show that these 'interactions' lead to phospholipid metabolism changes in ticks (see Figure 5). In the same vein, the subtitle on line 298 is misleading since the authors do not show that *B. burgdorferi* interacts with C1QL3.

2. Methods:

a. Lines 451-453 are out of place; they belong with the ISARL expression and binding experiments.

b. P25. How much adiponectin was injected per tick? How did the authors determine that was the appropriate amount?

c. Lines 543-546 appear to repeat information presented immediately above.

d. Lines 554-556. What concentration of C1QL3 was used for the binding experiments?

e. Figures 2a and b. More information is needed on how data in individual columns were combined and statistically analyzed.

3. Figures

a. In multiple figures, Y axes for spirochete burdens are incorrectly labeled as "ng flabB/1000 ng tick actin".

b. Figure 2a, b. What does each column represent (separate pools/biologic replicates, etc)? The degree of variability in some of the samples is concerning and probably explains why RT-PCR validation was poor. The hierarchical clustering seems to emphasize the lack of uniformity of results in like samples. It is confusing that the positions of dsISARL and dsGFP are switched in the two figures. The phylogenetic relationships in the heat maps have no value because the functional names of the genes aren't presented.

c. Figure 2E is just a list of functional categories. One cannot relate these categories to the genes in Figures 2a and b or the corresponding tables.

4. Referencing. There is a substantial literature that Lyme disease spirochetes in feeding ticks upregulate genes required for glycerol metabolism to exploit glycerol produced by the midgut. The Benach laboratory has elegantly demonstrated that Lyme disease spirochetes can extract lipids from the membranes of eukaryotic cells to which they are attached. None of this literature, which bolsters the findings reported herein, is cited.

*Reviewer #2 (Recommendations for the authors):*

Figure 4: Both assays are not very rigorous at the time to demonstrate interactions. It is recommendable to use a ligand binding assay, in a non-heterologous system such as tick guts cell culture.

Figure 5: The three little panels can be placed in figure 4 because they look like out of place in a figure by themselves.

Please revise statistics for figure S6A.

*Reviewer #3 (Recommendations for the authors):*

1) The 14-d skin samples give the initial appearance of a difference (Figure 1F). While it is stated in the text, Figure 1F should have statistical significance labeled on the figure. This will improve clarity and also make it consistent with the other figures.

2) I recommend bolding/starring or otherwise highlighting the ISARL gene in panels Figure 2 A/B.

---

## [Author Response]

Reviewer #1 (Recommendations for the authors):This is a technically impressive study that clearly breaks new ground in our understanding of the influence of Lyme disease spirochetes on the transcriptional/metabolic responses of tick midguts during an acquisition blood meal. The authors should address the following concerns:1. Textual clarity/interpretation of data:a. The ISARL knockdown RNAseq experiments (lines 163-201). The description of the results should be revised to better reflect the information in the corresponding tables (what is meant by "amino acid"). The two paragraphs on page 9 (lines 171-192) are particularly difficult to follow. One wonders what to make of these data given the fact that so many of the differentially regulated genes (13 of 17) identified by RNAseq were not confirmed by qRT-PCR (Figure S3). Also see comments on Figure 2 below.

Thank you for the comments. We revised this part to be more concise and only kept the critical and validated pathways (Page 9-10, Line 181-249). We made sure that our qPCR confirmation was rigorous with the testing of over ten biological replicates, which were totally independent of the sequencing samples (N=3). This increases the confidence that the significantly expressed genes in this study are regulated by ISARL upon *Borrelia* infection. We addressed the Figure 2 comments below.

b. It appears that mouse adiponectin injection experiments (p12) were done with flat ticks. Since midguts in flat ticks are metabolically quiescent, the rationale for this isn't clear. It is surprising that adiponectin effects in flat and engorged midguts are similar.

We agree that flat ticks are metabolically quiescent and different from engorging ticks. In addition, the process of injection into the ticks is an additional variable to consider. Despite these differences, the experiment demonstrates that adiponectin injection does induce alterations in the tick. We injected adiponectin protein into unfed ticks and waited for 8h for sample collection, to theoretically provide some time for a response to occur. Based on RNA-seq, adiponectin indeed stimulates 40 differently expressed genes (Supplementary file 1c). Since ticks are exposed to adiponectin present during a bloodmeal and the concentration of adiponectin in blood is relatively high (Avg: 3 μg/mL in mice, Figure 3H), it is possible that adiponectin can easily activate tick metabolism.

c. Lines 399-401 in the discussion. The authors do not demonstrate that B. burgdorferi interacts with C1QL3, nor do they show that these 'interactions' lead to phospholipid metabolism changes in ticks (see Figure 5). In the same vein, the subtitle on line 298 is misleading since the authors do not show that B. burgdorferi interacts with C1QL3.

We have added the data showing that after silencing *C1QL3*, *B. burgdorferi* did not significantly increase expression of *PTDSS1* in the nymphal tick guts (Page 16, Line 464). This is opposite to *PTDSS1* expression in naïve ticks upon *B. burgdorferi* infection (Figure 2I). We have also revised the subtitle as: C1QL3 is involved in the ISARL signaling pathway and modulates *B. burgdorferi* colonization.

In the Discussion, we have added that more studies need to be done to investigate the potential mechanisms of *B. burgdorferi* influence on C1QL3 expression (Page 19, Line 559-560) as suggested by reviewer 3.

2. Methods:a. Lines 451-453 are out of place; they belong with the ISARL expression and binding experiments.

We have moved this to the ISARL expression and binding experiment portion of the manuscript (Page 28, Line 759-761).

b. P25. How much adiponectin was injected per tick? How did the authors determine that was the appropriate amount?

This is a very good point. We were very careful when we injected the protein. We injected ~12 ng adiponectin per tick. We calculated the amount of protein according to the adiponectin concentration in murine blood (Avg: 3 μg/mL, Figure 3H) and in engorged nymphal ticks (Avg: ~4 mg). We have added this rationale into the revised manuscript (Page 27, Line 735-738).

c. Lines 543-546 appear to repeat information presented immediately above.

Thank you so much for your careful review. We have deleted this part.

d. Lines 554-556. What concentration of C1QL3 was used for the binding experiments?

We used 5μg rC1QL3 protein (0.5 μg/μL) for this experiment. We have provided this information in the revised manuscript (Page 29, Line 773).

e. Figures 2a and b. More information is needed on how data in individual columns were combined and statistically analyzed.

Thank you for your suggestion. Each column represents biologic replicates. We combined three biological replicates in each treatment for the differentially expressed gene analyses. We selected the significant genes for making the heatmaps. The significant genes were determined by the *p*-value and the adjusted *p*-value padj. All the *p*-values and padj values of the selected genes were far less than 0.05 in our study, and we have added the information in the revised manuscript (P28, Line 747-749).

3. Figuresa. In multiple figures, Y axes for spirochete burdens are incorrectly labeled as "ng flabB/1000 ng tick actin".

Thank you for your comments. We revised the labels as: flaB/actin ratio in all related figures.

b. Figure 2a, b. What does each column represent (separate pools/biologic replicates, etc)? The degree of variability in some of the samples is concerning and probably explains why RT-PCR validation was poor. The hierarchical clustering seems to emphasize the lack of uniformity of results in like samples. It is confusing that the positions of dsISARL and dsGFP are switched in the two figures. The phylogenetic relationships in the heat maps have no value because the functional names of the genes aren't presented.

Each column represents biologic replicates. Concerning the variability of biological triplicates in the RNA-seq, we included over ten biological replicates for RNA-seq validation by qPCR. All the samples were independent of the sequencing samples. Therefore, the significant expressed genes in this study are confirmed genes which are regulated by ISARL upon *Borrelia* infection.

We have revised the position of dsISARL and dsGFP, and kept them consistent in Figure 2a, 2b and 2c. We also presented gene names (as in Supplementary file 1a and 1b) instead of the Ixodes gene ID numbers in the heatmaps. In addition, we highlighted ISARL as suggested by Reviewer 3 in the revised manuscript.

c. Figure 2E is just a list of functional categories. One cannot relate these categories to the genes in Figures 2a and b or the corresponding tables.

We have deleted the Go function in the supplementary tables, which is the preliminary data analyzed in Uniprot. In the revised manuscript, we only kept the GO function and pathways based on DAVID analysis (Figure 2E).

4. Referencing. There is a substantial literature that Lyme disease spirochetes in feeding ticks upregulate genes required for glycerol metabolism to exploit glycerol produced by the midgut. The Benach laboratory has elegantly demonstrated that Lyme disease spirochetes can extract lipids from the membranes of eukaryotic cells to which they are attached. None of this literature, which bolsters the findings reported herein, is cited.

Thank you for this suggestion. We have added the information in the Discussion section (Page 19, Line 546-547) and cited the literature from Dr. Benach’s laboratory in the revised manuscript. We have also added the information that feeding ticks provide Lyme disease spirochetes with glycerol (Page19, Line 543-545). These data provide additional support for our findings.

Reviewer #2 (Recommendations for the authors):Figure 4: Both assays are not very rigorous at the time to demonstrate interactions. It is recommendable to use a ligand binding assay, in a non-heterologous system such as tick guts cell culture.

Thank you for your suggestion. Indeed, a non-heterologous data would be an additional validation for our study. We used tick ISE6 cells and found that C1QL3 indeed can bind to tick ISE6 cells. We have added this data in the revised manuscript (Page 16, Line 454-455 and Figure 4J).

Figure 5: The three little panels can be placed in figure 4 because they look like out of place in a figure by themselves.

We agree and have revised figure 4.

Please revise statistics for figure S6A.

Thank you for your comments. We double checked it and the statistics is correct.

Reviewer #3 (Recommendations for the authors):1) The 14-d skin samples give the initial appearance of a difference (Figure 1F). While it is stated in the text, Figure 1F should have statistical significance labeled on the figure. This will improve clarity and also make it consistent with the other figures.

There is no statistical significance in Figure 1F. We have now labeled it to be consistent with other figures. We also provide the raw data of statistics along with the revised manuscript.

2) I recommend bolding/starring or otherwise highlighting the ISARL gene in panels Figure 2 A/B.

Thank you for the good suggestion. We have provided the gene names as suggested by Reviewer 1, and highlighted ISARL gene with bold and red color accordingly.